# Enhancing Subjective Well-Being in Taiwanese University Students Through an Eight-Week Mindfulness-Based Program: A Pilot Study

**DOI:** 10.3390/bs14110980

**Published:** 2024-10-22

**Authors:** Li-Jen Lin, Su-Ping Yu, Yu-Hsun Lin, Yi-Lang Chen

**Affiliations:** 1General Education Center, Ming Chi University of Technology, New Taipei 243303, Taiwan; lljen@mail.mcut.edu.tw; 2Mindfulness Meditation Center, Ming Chi University of Technology, New Taipei 243303, Taiwan; spyu@mail.mcut.edu.tw (S.-P.Y.); yslin@mail.mcut.edu.tw (Y.-H.L.); 3Department of Industrial Engineering and Management, Ming Chi University of Technology, New Taipei 243303, Taiwan; 4Department of Business Administration, Ming Chi University of Technology, New Taipei 243303, Taiwan

**Keywords:** mindfulness, Oxford Happiness Questionnaire (OHQ), subjective well-being, course

## Abstract

While mindfulness has demonstrated extensive benefits across diverse populations, research on its integration as a formal course and its effects on subjective well-being (SWB), particularly among Taiwanese university students, remains limited. This pilot study examined changes in SWB among 85 Taiwanese university students (61 males, 24 females) following an 8-week Mindfulness-Based Cognitive Therapy for Life (MBCT-L) course. Participants included 38 lower-grade (pre-internship) and 47 senior (post-internship) students. SWB was measured using the 29-item Oxford Happiness Questionnaire (OHQ), rated on a 6-point Likert scale, and administered before and after the intervention. Results revealed a significant increase in students’ overall SWB scores from 3.69 to 3.92 post-intervention. Although females initially exhibited higher baseline SWB compared to males, no significant sex differences were observed after the course. Score discrepancies between pre- and post-test stages varied across sex and internship status, indicating differential impacts of the MBCT-L course on distinct demographic groups. Additionally, this study validated the reliability of the short version of the OHQ (8 items) for use in similar research contexts. By conducting a formal MBCT-L course for Taiwanese university students, this study provides valuable empirical evidence, contributing to the cross-cultural understanding of mindfulness-based interventions and their effects on SWB.

## 1. Introduction

Among the various approaches to enhancing psychological well-being, mindfulness has been shown to be effective [1,2,3,4,5]. Originally rooted in Buddhist traditions, mindfulness has been increasingly secularized and adapted for modern contexts, gaining significant attention as a potential intervention for improving mental health and well-being [6]. Mindfulness, defined as the state of being aware and attentive to present-moment experiences without judgment, has been integrated into various therapeutic approaches and wellness programs [7]. This includes both clinical interventions [8,9] and non-clinical applications [10,11], as well as implementation in real-world workplace settings [12,13,14].

Research conducted across various contexts and populations has shown that mindfulness-based interventions can effectively reduce stress, anxiety, and depressive symptoms while enhancing overall subjective well-being (SWB) [15,16,17]. However, some studies have reported no significant association between mindfulness training and well-being [18,19]. These findings suggest that mindfulness training programs may need to emphasize cultivating a non-judgmental and accepting attitude toward one’s thoughts and feelings to effectively enhance happiness [19].

Given the numerous theories on the study of well-being, the measurement tools developed based on these theories also vary. One widely used instrument for assessing psychological well-being is the Oxford Happiness Questionnaire (OHQ), created by Hills and Argyle [20]. It was primarily developed based on the Oxford Happiness Index (OHI) [21]. Happiness is often considered an important component of SWB. Therefore, the OHQ can indirectly reflect SWB, though it focuses more specifically on measuring happiness itself. The OHQ consists of 29 items and is designed without distinct subdivisions into multiple constructs or subscales, unlike some other psychological measurement tools [20]. Although some studies have attempted to perform factor analyses on the OHQ to identify potential dimensions, the results have generally been inconsistent [22,23,24]. Additionally, the original designers of the questionnaire did not explicitly define the dimensions it encompasses. Consequently, the OHQ is typically regarded as a unidimensional scale for happiness, used to reflect an individual’s overall level of happiness. Hills and Argyle [20] also developed a shortened version of the OHQ to enhance evaluation efficiency. This abbreviated instrument consists of 8 items selected from the original 29-item scale. Studies have demonstrated that the shorter version retains validity comparable to the full OHQ [20,25], while significantly reducing administration time.

While mindfulness-based interventions have been extensively studied worldwide, there is increasing interest in understanding their effectiveness across different cultural contexts [26,27,28,29]. Taiwan, with its unique blend of traditional Eastern philosophies and modern influences, provides a compelling context for exploring the impact of mindfulness practices. Despite this cultural relevance, research on structured mindfulness programs among Taiwanese university students remains limited. Previous mindfulness interventions have often relied on online [30,31,32,33], short-term [34,35,36,37], or informal training methods [38,39,40,41]. Given the increasing academic and life pressures faced by university students today, incorporating mindfulness into formal university courses presents an important area of exploration.

In recent years, there has been increasing concern about the mental health and psychological well-being of university students worldwide [42]. The transition to higher education, coupled with academic pressures, social challenges, and uncertainties about the future, often contributes to elevated levels of stress, anxiety, and depression among this population [43,44]. Research indicates that Taiwanese university students face comparable challenges, with a significant number reporting symptoms of psychological distress [45,46,47]. University life, marked by unique demands and increased autonomy, can exacerbate mental health difficulties and lead to maladaptive behaviors in some students [27]. Additionally, various studies have demonstrated that university students at different academic levels experience distinct patterns of stress and anxiety, with notable differences observed between freshmen [47] and senior students [43]. Given that these students represent the future leaders of society, enhancing their well-being through targeted interventions and course implementation has become a critical priority [48].

Meta-analyses have demonstrated that various mindfulness-based interventions, including Mindfulness-Based Cognitive Therapy (MBCT), can significantly improve mental health outcomes. However, research on the specific impact of MBCT for Life (MBCT-L) on SWB remains limited [11]. Although traditional MBCT has been more widely implemented, the adapted MBCT-L program has seen more restricted use, primarily in specific populations such as healthcare staff [14], teachers [49], and participants in psychedelic-assisted psychotherapy [50]. While MBCT-L is particularly suited to non-clinical populations, including workplace environments and student populations, studies investigating its application among university students are still scarce.

This study aims to address the gap in mindfulness research within Taiwan by examining the effects of an eight-week formal mindfulness-based course on the SWB of Taiwanese university students. By adapting the established MBCT-L protocol [51] to the local context and utilizing the full version of the OHQ (29 items, as detailed in the Appendix A) [20] to assess SWB, we contribute to the growing body of cross-cultural research on mindfulness interventions. This pilot study provides valuable insights into the feasibility and acceptability of implementing such programs within Taiwan’s higher education system and explores the influences of the MBCT-L intervention on factors such as sex and internship experience. Additionally, this study aimed to investigate the efficacy of the abbreviated OHQ (8 items) in assessing SWB among Taiwanese university students, contributing to the validation of this instrument in a specific cultural context. We hypothesized that the introduced formal mindfulness course would positively influence SWB and that individual factors, such as sex and internship experience, would lead to varying effects due to their potential relationship with SWB. We also hypothesized that the shorter version of the OHQ would demonstrate comparable validity to the full version in assessing SWB among Taiwanese university students. Confirming this hypothesis would have important implications for future research, enabling more efficient and time-effective assessments of SWB in this population.

## 2. Materials and Methods

### 2.1. Participants

Participants were 87 undergraduates (61 males and 26 females) aged 19 to 23 years, enrolled during the fall of 2023 at Ming Chi University of Technology (MCUT), New Taipei, Taiwan. They represented a general student population, with no pre-screening for individuals with high care needs (e.g., depression or bipolar disorder). None of the participants had received any prior mindfulness-based intervention or course before this study. All participants voluntarily enrolled in the “Mindfulness, Well-Being, and Creativity” elective course offered by the General Education Center at MCUT.

To facilitate effective course delivery, the 87 students were randomly divided into three evenly sized classes. Two female participants were excluded during the data collection phase due to not meeting the study’s criteria, resulting in a final sample of 61 males and 24 females, for a total of 85 students. MCUT is a well-known institution in Taiwan, specializing in engineering and technical education. The gender distribution of participants in this study, with a male-to-female ratio of approximately 2.5:1, closely mirrored the overall gender composition of the university. A priori power analysis was conducted using G*Power (Version 3.1.9.7) for the *t*-test, despite the sample size being limited by course enrollment. The analysis assumed a power of (1 − β) = 0.80, an α-error probability of 0.05, and a medium effect size of 0.35. Results indicated a required sample size of 67 subjects. Our actual sample of 85 participants thus exceeded the minimum requirement for statistical power.

The study focused on the mindfulness component of a 16-week course that combined mindfulness and creativity training. This comprehensive course was conducted for two hours weekly, with the initial eight weeks dedicated to the mindfulness sub-course that formed the primary focus of this investigation. Among the participants, 20 were first-year students, 17 were second-year students, and 48 were fourth-year students. At MCUT, juniors are required to participate in a full year of off-campus internships, so no third-year students were present on campus. For this study, the 37 students in the first and second years were categorized as the lower-grade group (having no internship experience), while the fourth-year students, who had completed one year of internships, were categorized as the senior group. All participants provided informed consent, and the experiment adhered to the 2013 World Medical Association Declaration of Helsinki. The study received ethical approval from the Ethics Committee of Chang Gung Memorial Hospital, Taiwan (code: 202202334B0).

### 2.2. Mindfulness-Based Course

In our study, the mindfulness-based course was designed using the MBCT-L framework [48,49]. The course was facilitated by Dr. L.-J. Lin, the first author, who holds certifications in MBCT-L Teacher Training (awarded in October 2020) and as an M1-M4 Team Supervisor (from April 2016 to March 2019) from the Mindfulness Centre at the University of Oxford. The MBCT-L course, derived from the original MBCT program [52], was designed by the Mindfulness Centre to be accessible to the general population, aiming to enhance mental health and well-being in everyday settings [53]. MBCT, which integrates cognitive behavioral therapy (CBT) within a mindfulness-based framework, was initially developed to prevent depressive relapse, a context in which it has demonstrated significant benefits [54]. MBCT-L is an adaptation of MBCT for non-clinical populations, maintaining the core structure and techniques while placing greater emphasis on well-being, appreciation, and gratitude.

The MBCT-L course was delivered as an 8-week group intervention, preceded by a pre-course orientation session. During this orientation, participants were introduced to mindfulness practices and engaged with various CBT strategies. In this study, each group was led by the same instructor, Dr. L-J. Lin, with a maximum of 30 students per class. The course was structured around weekly themes, each focusing on a key aspect of mindfulness. These themes included mindfulness-based stress reduction, enhancing mind–body awareness, recognizing thoughts as mental events rather than facts, and integrating mindfulness into everyday life. The instructor guided participants through a range of formal and informal mindfulness practices, such as meditation, mindful breathing, body scans, and gratitude exercises. Each 2 h session was supplemented by daily home practice, which students were encouraged to maintain for approximately 40 min [55]. During the following sessions, students were invited to share their experiences, helping to create a supportive learning environment. A summary of the 8-week course structure is provided below, with further details available on the official webpage: https://www.mbct.life/programme (accessed on 8 September 2024).

■Guided instruction in mindfulness meditation.■Gentle stretching and movement, some of which could be done on the floor.■Group dialogue and discussions focused on enhancing awareness in everyday life.■Daily home assignments to reinforce the practices and concepts learned in each session.

### 2.3. Oxford Happiness Questionnaire (OHQ)

The OHQ was developed as an enhanced version of the OHI [20]. In this study, we utilized the OHQ (see Appendix A), which comprises 29 items presented as individual statements. Each item is rated on a 6-point Likert scale, ranging from 1 (strongly disagree) to 6 (strongly agree). The revised OHQ is more streamlined and easier to administer than the OHI, offering a wider range of response options. It has demonstrated strong reliability (α = 0.91) and satisfactory validity, showing stronger correlations with various personality traits associated with well-being [20].

Given that this study took place in Taiwan, the questionnaire was administered in Chinese. The Chinese version of the OHQ has also demonstrated strong reliability and validity [56,57]. The course instructor provided participants with a detailed explanation of the study’s purpose and the relevant concepts before they completed the questionnaire. Participants filled out the paper-and-pencil questionnaire at both the pre-test and post-test stages, before and after the MBCT-L intervention.

This study also assessed outcomes using a shortened version of the OHQ. The abbreviated scale, consisting of 8 items from the original 29-item instrument [20], is commonly used to evaluate SWB [25,58]. The short version includes items 1, 3, 12, 13, 16, 18, 21, and 29, as outlined in the Appendix A. In the current study, internal consistency was examined for both the full and short versions of the OHQ. The Cronbach’s α coefficients were 0.933 for the full version and 0.758 for the short version, indicating excellent and acceptable internal consistency, respectively.

### 2.4. Procedure

The mindfulness intervention for this study was delivered through a formal course provided by the General Education Center at MCUT. The course, designed as an elective, was available to undergraduate students from all years (1–4), with each class limited to fewer than 30 participants. Following enrollment, the course instructor outlined the study’s objectives and methods, giving students the option to opt out if they wished. Ultimately, 87 students were enrolled across three classes, and 85 completed the entire course.

The pre-test OHQ questionnaire was administered during the first week of the course to serve as a baseline for the MBCT-L intervention. Immediately following the eight-week MBCT-L course, a post-test of the OHQ was conducted, and the results were compared with the pre-test to evaluate the impact of the intervention. Throughout the mindfulness course, students were required to maintain a regular schedule and avoid major life events, such as long-distance travel or significant career decisions, to ensure consistency. All interventions were conducted in a realistic classroom setting.

### 2.5. Statistical Analysis

The data for this study were analyzed using SPSS 23.0 statistical software (IBM Corp., Armonk, NY, USA), with a significance level set at 0.05 for all statistical tests. To ensure analysis robustness, the Shapiro–Wilk test assessed the normal distribution of numerical variables, and Levene’s test gauged homogeneity of variances. The primary goal was to assess the impact of the eight-week MBCT-L course on students’ SWB scores. Additionally, the study examined how sex (male and female) and internship experience (lower grade students and senior students) influenced SWB scores. An independent *t*-test was used to compare differences between groups, while a paired *t*-test assessed differences within groups before and after the intervention. Additionally, Pearson product-moment correlation analysis was conducted to explore the relationship between the full and short versions of the OHQ. In this study, first-year (n = 20) and second-year students (n = 18) were combined into a single lower-grade group (n = 38). To ensure that there were no significant differences in OHQ scores between these two cohorts, independent *t*-tests were conducted. This analysis was performed to confirm that the data from the two academic years could be aggregated without biasing the results.

## 3. Results

The Shapiro–Wilk test confirmed that the data analyzed in this study followed normal distributions, and Levene’s test verified the homogeneity of variances, ensuring that the necessary assumptions for subsequent analyses were met. Before conducting the main analysis, potential differences in relevant traits between first- and second-year university students were examined. Comparisons of OHQ scores across all 29 items showed no significant differences between these two cohorts. The t-values for pre-test scores ranged from −1.473 to 1.624, and for post-test scores from −1.568 to 1.043, with all *p*-values exceeding 0.05.

The descriptive statistics and independent *t*-test results comparing the pre- and post-test scores of the 85 students for each item of the OHQ are shown in Table 1. Among the 29 items, 11 exhibited significant differences. Overall, the mean (standard deviation) scores before and after the MBCT-L course intervention were 3.69 (0.68) and 3.92 (0.80), respectively, indicating a significant difference (*p* < 0.001) and demonstrating the effectiveness of the mindfulness-based intervention in improving specific aspects of psychological well-being.

Table 2 highlights the significant differences in OHQ item scores between the pre-test and post-test for each specific group (males vs. females, lower grades vs. senior grades). Only items showing significant differences are presented, demonstrating the varied impact of the mindfulness-based course on different demographic groups. Independent *t*-tests revealed significant differences in total OHQ scores between the two test stages across all subgroups (*p* < 0.05).

When comparing the pre- and post-test OHQ scores between different paired groups (sex or internship), it was observed that before the mindfulness course intervention, female students had significantly higher scores (3.90) compared to male students (3.62) (*p* < 0.05), as shown in Figure 1. However, after the intervention, there was no significant difference between the two groups (4.09 vs. 3.85), suggesting that women generally have higher baseline levels of SWB compared to men. Regarding internship experience, there were no significant differences in OHQ scores between students with or without internship experience in both the pre-test and post-test phases (Figure 2).

In our analyses, the correlation coefficients between the average scores of the full (29 items) and short versions of the OHQ (8 items) in the pre-test and post-test were 0.825 (*p* < 0.001) and 0.903 (*p* < 0.001), respectively. Figure 3 illustrates the correlations between these two versions of the OHQ in the pre- and post-test stages, respectively, for all 85 participating students.

## 4. Discussion

Mindfulness-based interventions have consistently demonstrated effectiveness in enhancing psychological well-being. However, previous research has often relied on informal, online, or short-term approaches. This study, in contrast, examined a formal university course designed for a particularly stressed population—university students. The goal was to evaluate the impact of an eight-week mindfulness-based elective course on students’ SWB. As expected, the MBCT-L course resulted in a significant improvement in SWB, with OHQ scores increasing from 3.69 to 3.92 (*p* < 0.001). This result underscores the intervention’s positive effects on stress and anxiety reduction among the students, aligning with prior research on the broad benefits of mindfulness interventions across various settings [1,2,3,4,5]. By focusing on Taiwanese university students, this study adds valuable local empirical data to the cross-cultural understanding of mindfulness-based interventions [26,28,29].

When compared to similar studies, our findings show both consistency and unique contributions. For instance, Felver et al. [59] reported significant reductions in psychological distress among college students following a mindfulness-based stress reduction (MBSR) program, which shares conceptual similarities with our MBCT-L intervention. Although their study used different outcome measures, the effect sizes in improving overall well-being were comparable. Additionally, Tran et al. [60] highlighted the role of mindfulness in enhancing self-compassion and psychological well-being while mitigating anxiety, depression, and stress, particularly during the COVID-19 pandemic. Ortet et al. [19] emphasized the importance of a non-judgmental and accepting attitude in mindfulness training to improve happiness, elements that appear to have been successfully integrated into our MBCT-L course. This is evidenced by the significant improvements in self-acceptance and life satisfaction, as measured by the OHQ. Moreover, post-course evaluations conducted through MCUT’s routine feedback process revealed high levels of student satisfaction, with participants assigning the course an overall rating of 93 out of 100, indicating strong approval of the program’s content and delivery.

The analysis revealed significant improvements in 11 of the 29 OHQ items, demonstrating that the mindfulness-based course positively affected specific aspects of well-being. Notable improvements were seen in areas such as life satisfaction, interpersonal warmth, optimism, and self-perception. For example, item 1 (“I don’t feel particularly pleased with the way I am”) showed a significant increase from 2.88 to 3.60 (*p* < 0.001), reflecting enhanced self-acceptance. Similarly, item 3 (“I feel that life is very rewarding”) increased from 3.69 to 4.18 (*p* < 0.01), indicating a greater appreciation for life. These results align with Khoury et al. [15], who conducted a meta-analysis of mindfulness-based therapy and reported significant improvements in various psychological outcomes, including quality of life and SWB.

The absence of increases in certain OHQ items does not necessarily indicate that the intervention was ineffective. It is more likely that some items already had high baseline scores, such as items 8, 10, 11, 16, and 29, which may reflect life circumstances particularly relevant to university students, especially those at MCUT. These items correspond to well-established concepts from the previous literature, including being committed and involved [61], believing the world is good [62,63], finding beauty in things [64,65], laughing a lot [66], and having happy memories [67]. The clear connection between these items and current SWB, as well as their relevance to students, may merit further investigation.

Interestingly, the study found variations in the effects of the mindfulness course based on sex and internship experience. Female students initially reported higher baseline levels of SWB compared to their male peers. Although most research shows only slight differences in happiness and life satisfaction between men and women, some studies suggest that women, particularly younger women, report higher SWB than men [68]. For example, Inglehart [69] identified significant gender-related differences in SWB, which are often obscured by the interplay of age, gender, and well-being. Younger women generally report higher levels of happiness than younger men, while older women tend to report lower happiness levels. Similarly, research in China indicates that women generally exhibit higher well-being than men, although factors like the burden of housework can negatively affect this [70]. Our study focused on a sample of young, unmarried women. Notably, Rojiani et al. [71] also found that women experienced greater reductions in anxiety and negative affect, along with increased mindfulness, following an intervention. However, in our study, post-intervention results showed no significant differences in SWB between male and female students, suggesting that MBCT-L may help reduce gender disparities in SWB among university students.

Regarding internship experience, while no statistically significant differences in OHQ scores were found between students with and without internship experience in both pre-test and post-test phases, distinct patterns emerged in specific items for each group (Table 2). This suggests that individual differences and life experiences are important factors to consider when implementing mindfulness interventions. Although overall differences were not statistically significant, senior students had slightly lower SWB scores than lower-grade students at the post-test stage (3.85 vs. 3.99). This may reflect increased anxiety about future employment as students near graduation [72,73], despite gaining valuable work experience through internships [74]. This finding underscores the potential need for targeted psychological counseling for students as they near graduation. Few studies have specifically examined the role of internship experience in mindfulness outcomes, making our findings a valuable addition to the literature.

The strong correlations between the full 29-item and the short 8-item versions of the OHQ in both the pre-test (r = 0.825, *p* < 0.001) and post-test (r = 0.903, *p* < 0.001) stages suggest that the short version is a reliable and efficient tool for assessing SWB in similar research contexts, as shown in Figure 3. This has practical implications for future studies, as it supports the use of quicker assessments without compromising data quality. The consistency of these results with those observed in Western participants [20,75] further suggests that the short version of the OHQ is applicable to Taiwanese university students.

Despite the promising findings, this study has several limitations that should be acknowledged. First, the absence of a control group limits our ability to definitively attribute the observed improvements in well-being to the mindfulness intervention. Future research should address this by employing randomized controlled designs. Second, the relatively small sample size (n = 85) from a single Taiwanese university may restrict the generalizability of the findings to broader populations or different cultural contexts. Additionally, due to sample size constraints related to elective course enrollment, freshmen and sophomores were combined into a single lower-grade cohort. Although statistical analyses showed no significant differences in OHQ scores between these two academic years, a larger sample that includes a wider range of grade levels would allow for more nuanced analyses of grade-specific effects. Third, the students who enrolled in the mindfulness course may have had a pre-existing interest in mindfulness, which could introduce a pre-screening bias. To confirm the effectiveness of MBCT-L across diverse student groups, larger-scale, multi-site studies are necessary. Furthermore, the study relied solely on self-report measures of well-being. While the OHQ is a validated tool, incorporating objective measures or qualitative data could provide a more comprehensive understanding of the intervention’s impact. Lastly, the study did not include a follow-up assessment to evaluate the long-term effects of the mindfulness course. Future research should employ longitudinal designs to assess the sustainability of improvements in SWB over time.

## 5. Conclusions

This pilot study offers compelling evidence of the positive impact of an eight-week formal MBCT-L course on the SWB of Taiwanese university students. The significant increase in overall OHQ scores and improvements across specific well-being dimensions indicate that incorporating mindfulness training into formal university curricula could be an effective strategy for enhancing student mental health. These findings add to cross-cultural research on mindfulness-based interventions, highlighting their relevance in the Taiwanese context. From an intervention perspective, the results suggest that MBCT-L could be systematically integrated into higher education programs in Taiwan through customized curricula, large-scale longitudinal studies, and targeted faculty training. The observed differences in outcomes based on sex and internship experience underscore the need for flexible and personalized approaches when implementing mindfulness programs. Additionally, the strong correlation between the full and short OHQ versions supports the use of the abbreviated version in future research, offering practical benefits for streamlining assessments in larger-scale studies. As universities place increasing emphasis on student well-being, this study provides a foundation for culturally sensitive, evidence-based mindfulness programs that can be adapted and scaled to address the diverse needs of student populations.

## Figures and Tables

**Figure 1 behavsci-14-00980-f001:**
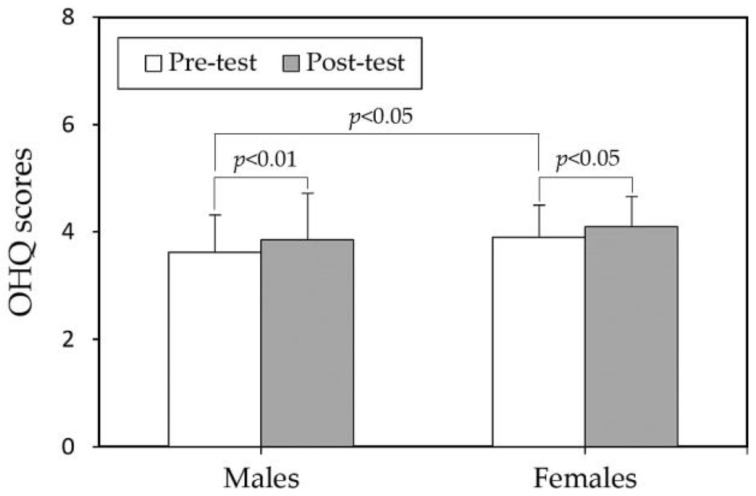
Comparisons of Oxford Happiness Questionnaire scores (OHQ) between sexes in both test stages.

**Figure 2 behavsci-14-00980-f002:**
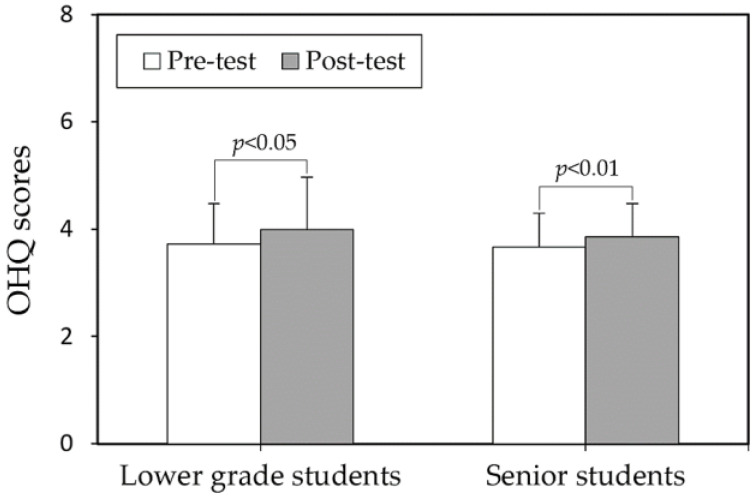
Comparisons of Oxford Happiness Questionnaire scores (OHQ) between internship experience levels in both test stages.

**Figure 3 behavsci-14-00980-f003:**
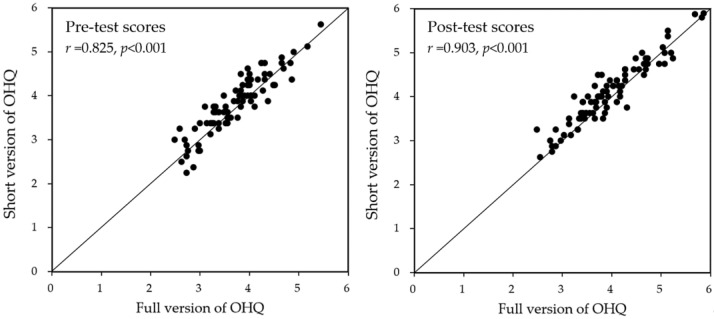
The correlations between the two Oxford Happiness Questionnaire (OHQ) versions in both pre- and post-test stages.

**Table 1 behavsci-14-00980-t001:** The paired *t*-test results for the 29 items of the Oxford Happiness Questionnaire.

Items	Pre-Test				Post-Test				Mean Difference	t	*p*
	Mean	SD	γ_1_	γ_2_	Mean	SD	γ_1_	γ_2_			
1	2.88	1.24	0.18	−0.35	3.60	1.50	−0.04	−1.06	0.72	4.275	<0.001
2	3.74	1.22	−0.17	−0.63	3.73	1.19	−0.04	−0.25	−0.01	−0.083	0.934
3	3.69	1.25	−0.22	−0.43	4.18	1.23	−0.11	0.08	0.49	2.873	<0.01
4	3.45	1.25	0.06	−0.37	3.89	1.13	−0.21	0.02	0.44	3.054	<0.01
5	2.64	1.25	0.20	−0.43	2.66	1.31	0.22	−0.23	0.02	0.141	0.888
6	3.53	1.28	0.24	−0.57	3.80	1.51	−0.06	−1.08	0.27	2.015	<0.05
7	4.08	1.23	−0.16	0.20	4.40	1.15	−0.20	0.34	0.32	2.652	<0.01
8	4.14	1.01	−0.23	0.32	4.26	1.20	−0.22	−0.47	0.12	0.844	0.401
9	4.01	1.28	−0.20	−0.30	4.31	1.24	−0.25	0.23	0.30	2.418	<0.05
10	4.22	1.42	−0.21	−0.46	4.35	1.35	−0.20	−0.29	0.13	0.926	0.357
11	4.19	1.36	−0.23	−0.01	4.28	1.32	−0.17	0.14	0.09	0.741	0.461
12	3.68	1.17	−0.19	0.14	3.99	1.22	−0.20	−0.24	0.31	2.340	<0.05
13	3.21	1.25	0.20	−0.15	3.67	1.32	0.17	−0.81	0.46	3.645	<0.001
14	2.82	1.17	0.16	−0.19	3.02	1.25	0.23	0.05	0.20	1.179	0.242
15	3.56	1.30	−0.16	−0.48	3.81	1.29	−0.14	−0.22	0.25	1.974	0.052
16	4.22	1.18	−0.22	1.02	4.34	0.98	0.02	−0.38	0.12	1.106	0.272
17	3.78	1.10	−0.14	0.71	4.06	1.15	−0.19	0.01	0.28	2.371	<0.05
18	4.04	1.13	−0.13	0.74	4.06	1.07	−0.05	−0.14	0.02	0.191	0.849
19	3.71	1.22	−0.01	−0.34	3.91	1.24	−0.05	−0.19	0.20	1.221	0.226
20	3.76	1.20	0.04	−0.63	3.85	1.13	0.07	−0.59	0.09	0.701	0.485
21	3.56	1.20	0.02	−0.24	3.68	1.14	−0.17	0.13	0.12	0.856	0.394
22	3.94	1.15	−0.12	−0.14	4.05	1.18	−0.21	0.48	0.11	0.817	0.416
23	2.95	1.23	0.20	−0.09	3.24	1.39	0.21	−0.55	0.29	1.940	0.056
24	3.62	1.61	0.01	−1.19	3.92	1.55	−0.04	−1.06	0.30	1.824	0.072
25	3.65	1.30	−0.18	−0.14	3.71	1.16	0.07	−0.62	0.06	0.540	0.591
26	3.24	1.00	−0.19	−0.26	3.48	1.12	0.11	−0.20	0.24	1.778	0.079
27	4.33	1.08	−0.18	0.98	4.56	1.05	−0.15	−0.62	0.23	2.178	<0.05
28	3.40	1.47	0.15	−0.78	3.76	1.41	−0.18	−0.66	0.36	2.338	<0.05
29	5.09	1.16	−0.22	1.20	4.95	1.19	−0.21	0.51	−0.14	−1.053	0.295
Total	3.69	0.68	0.16	0.89	3.92	0.80	−0.24	1.34	0.23	3.306	<0.001

Note: All data range from 1 (minimum) to 6 (maximum) for each item; SD, standard deviation; γ_1_, skewness; γ_2_, kurtosis.

**Table 2 behavsci-14-00980-t002:** Significant item score differences in the Oxford Happiness Questionnaire for each group using independent *t*-test.

	Student Sex		Internship	
Items	Males (n = 61)	Females (n = 24)	Lower Grades (n = 38)	Senior Grades (n = 47)
Pleased with self	2.95/3.61 ***	2.71/3.58 **	2.89/3.55 *	2.87/3.64 ***
2. Interested in others	―	―	―	―
3. Life is rewarding	3.54/4.00 *	4.08/4.63 **	3.50/4.18 **	―
4. Warmth for others	3.36/3.77 *	3.67/4.21 *	3.47/4.05 *	3.43/3.77 **
5. Wake up rested	―	―	―	―
6. Optimistic	3.52/3.82 *	―	―	―
7. Find things amusing	3.87/4.26 **	―	3.92/4.50 **	―
8. Committed and involved	―	―	―	―
9. Life is good	3.84/4.15 *	―	3.97/4.58 **	―
10. World is good	―	―	―	―
11. Laugh a lot	―	―	―	―
12. Satisfied with life	3.61/3.92 *	―	3.58/4.24 **	―
13. Look attractive	3.18/3.69 ***	―	3.08/3.63 **	3.32/3.70 **
14. Can organize time	―	―	―	―
15. Feel happy	3.44/3.75 *	―	3.58/4.18 **	―
16. Find beauty in things	―	―	―	―
17. Cheerful effect on others	3.62/3.92 *	―	―	―
18. Done things wanted	―	―	―	―
19. In control	―	―	―	―
20. Can do most things	―	―	―	―
21. Mentally alert	3.48/3.77 *	―	―	―
22. Joy and elation	―	―	―	―
23. Make decision easily	2.97/3.30 *	―	―	―
24. Life has meaning and purpose	2.97/3.80 ***	―	―	―
25. Feel energetic	―	―	―	―
26. Good influences	―	3.21/3.54 *	3.26/3.71 *	―
27. Have fun with others	4.23/4.52 **	―	―	4.17/4.49 *
28. Feel healthy	3.48/3.79 *	―	―	3.23/3.72 **
29. Happy memories	―	―	―	―
Total	3.62/3.85 **	3.89/4.09 *	3.72/3.99 *	3.67/3.85 **

Note: The item description refers to the study of Hills and Argyle [20] and data are presented as pre-test score/post-test score; * *p* < 0.05, ** *p* < 0.01, *** *p* < 0.001.

## Data Availability

The data are available upon reasonable request to the Corresponding Author.

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
