# Peer review of "Enhancing Subjective Well-Being in Taiwanese University Students Through an Eight-Week Mindfulness-Based Program: A Pilot Study"

_behavsci, 2024, doi:10.3390/bs14110980_

Round 1

Reviewer 1 Report

Comments and Suggestions for Authors

1. It is suggested that using the short version is reliable because there is a high correlation between the long version and short version of the assessment tool. Is the necessity of this research question presented in the introduction? Isn't this more of a necessary procedure in selecting the version of the measurement tool in the research process, rather than a research result itself?

2. When the dependent variable follows a normal distribution, an assumption that a paired t-test could be conducted would be required. Thus, it is necessary to present the results of verifying this aspect.

3. Looking at previous studies (i.e., life is rewarding, mentally alert, pleased with self, find beauty in things, satisfied with life, can oarganise time, look attractive, happy menories)(Hills, P., & Argyle, M. (2002)), it is confirmed that the OHQ (Oxford Happiness Questionnaire) has 8 factors. In this study, after going through the factor analysis procedure, why wasn't an independent t-test conducted on the results of the factor analysis for the 30 items?

4. In the descriptive statistics, please present the minimum, maximum, skewness, and kurtosis.

Reviewer 2 Report

Comments and Suggestions for Authors

it was a well written manuscript. congratulations to authors. However, there is a need for improvement.

1. why female students participation is less in the study?

2. As there was students from different years, kindly present a table where results are shown year wise. 

3. Who are more stressed first years or final years?

4. Cite latest references

5. Have you collected feedback from the students about the program?

Comments on the Quality of English Language

please check the grammatic errors in discussion section

Reviewer 3 Report

Comments and Suggestions for Authors

Thank you very much for giving me the opportunity to review this interesting paper entitled Enhancing Subjective Well-being in Taiwanese University Students Through an Eight-Week Mindfulness-Based Program: A Pilot Study. Altogether it is an interesting and well written article that its main aim was to investigate changes in subjective wellbeing (SWB) among 85 Taiwanese university students (38 lower-grade students and 47 senior students) following an 8-week Mindfulness-based Cognitive Therapy for Life (MBCT-L). As authors noted despite the extensive benefits of mindfulness on subjective wellbeing across diverse populations are well-documented, actual interventions remain limited, particularly among Taiwanese university students. Their results obtained in this study revealed that the students' overall SWB scores significantly increased after the intervention.

In general, the manuscript is clear, and relevant for the field, in fact as the authors noted the research around this topic is scarce, especially in this geographical context. However, it would be interesting to present it in a more well-structured manner.  The cited references are mostly recent publications, however, if possible, it more could be added references from 2024. There are no arguments to the adequacy of the sample size regarding the analyses performed. Authors maybe could test it using, for example, gpower. The description of the intervention could be more detailled. The figures and tables are appropriate. Overall, in my humble opinion, it is a study that can be useful for the field, and it would be interesting to introduce some improvements that would enhance the value of the results obtained.

Say this, I have some comments: 

1.           Perhaps it would be interesting to rethink your abstract. It seems that the objective of your study is to evaluate the effect of an intervention with mindfulness on Taiwanese university students’ well-being (in fact, in lines 188-189 you said this is your primary objective). However, your abstract advances later (in lines 28-31) what seems to be another added objective: to estimate whether the two measures that are indicated (Oxford Happiness Questionnaire (OHQ) and its short version) are equally valid for assessing psychological well-being. Moreover, I am not sure due to the length of words suggested by the journal (around 200) if it is necessary to offer t and p values in the abstract.

2.           In this vein, despite the introduction is well written, in my modest opinion it would be interesting to rethink it. Please help the reader by perhaps restructuring a bit the way in which the information is provided, by being more concrete (e.g. by providing some example to better contextualize your statement ‘Although meta-analyses have shown that various mindfulness based interventions can improve mental health there is a notable lack of discussion on the specific impact of MBCT-L on SWB’) and by outlining your objective in more detail.

3.           In the materials and methods section, when describing your measurements it can be confusing for the reader whether you are referring to the long version of the Oxford Happiness Questionnaire (OHQ) or the short version, for example in line 180, the reliability value you give is that of the long version or the short version? And if it is the short version, why don't you offer this value in the lines 162-169 where you describe it? Why these eight items exactly for this short version?

4.           In your conclusions section, perhaps you could reinforce more what your work entails from an interventional and practical point of view for future developments.
